# Variability in the Agronomic Behavior of 12 White Grapevine Varieties Grown under Severe Water Stress Conditions in the La Mancha Wine Region

A. Sergio Serrano [1,2], Jesús Martínez-Gascueña [1], Gonzalo L. Alonso [2], Cristina Cebrián-Tarancón [2], M. Dolores Carmona [1], Adela Mena Morales [1] and Juan L. Chacón-Vozmediano [1,*]

[1] Regional Institute of Agri-Food and Forestry Research and Development of Castilla-La Mancha (IRIAF), Ctra. Toledo-Albacete s/n, 13700 Tomelloso, Spain
[2] Departament of Agricultural Chemistry, School of Agricultural and Forestry Engineering and Biotechnology, University of Castilla-La Mancha, Avda. de España s/n, 02071 Albacete, Spain
* Correspondence: jlchacon@jccm.es

**Abstract:** Viticulture around the world is currently affected by climate change, which is causing an increasing scarcity of water resources necessary for the maintenance of vineyards. Despite the drought hardiness of grapevine (*Vitis vinifera* L.), this threat seriously compromises its cultivation in the near future, particularly in wine-growing areas with a semi-arid climate. Identifying varieties capable of producing suitable yields and good-quality grapes under drought conditions is integral to ensuring the sustainability of the wine sector. This study focuses on vines from both minority and widely grown varieties, which were supplied only with the water intended to ensure their survival. The carbon and oxygen isotope ratios, yield, and quality parameters were evaluated on the vines and musts during the period of 2018–2020. The results revealed that not all varieties responded equally well to drought. Albillo Real, Coloraillo, Macabeo, and Verdejo adapted well to drought conditions, simultaneously maintaining high yields and must quality. By contrast, Pedro Ximénez can be considered poorly adapted. This variety was the one that produced the lowest yield and had low acidity levels in the must.

**Keywords:** carbon isotope ratio; drought; must quality; oxygen isotope ratio; yield





## 1. Introduction

The future of viticulture is seriously threatened by climate change, and questions remain about how it will evolve over the coming years [1]. In recent decades, the scarcity of water resources has intensified, precipitation patterns have changed, and the frequency of extreme events—such as droughts and heat waves—has increased, thereby affecting viticulture [2–4]. In grapevines, this set of adverse phenomena leads to a deterioration of functional plant mechanisms, affecting growth, physiology, and grape ripening, which may cause severe losses with respect to yield and quality [5]. Under water stress conditions, vegetative development and vine yield components have been demonstrated to be significantly affected, particularly the berry size [6–8]. Water stress also influences the number of bunches that develop inside the buds and which will be exhibited during the following vegetative period [9]. However, a water deficit can improve grape quality unless it is severe [10]. When drought is also associated with high temperatures, it can unbalance the chemical composition of the grape, resulting in overripe grapes with low acidity and a high sugar content [11–14].

Currently, the exploitation of water resources is considered an unsustainable practice in the long term; indeed, it may not be allowed or may be severely limited in the future through legislative regulation. These restrictions could make vine cultivation in the Mediterranean region infeasible or at least unsustainable in this century [1]. Therefore, new adaptation

strategies toward sustainable viticulture should be explored to maintain grape yield and quality [14–16]. In this scenario, the use of drought-tolerant varieties could be considered an interesting alternative [3,17], particularly in semi-arid areas where water resources are increasingly limited [18,19]. In this regard, the high genetic diversity of grapevines represents an advantage as growers seek to identify the varieties with the most beneficial drought-tolerant traits [3].

Stable isotope ratios, primarily of carbon ($\delta^{13}$C) and, to a lesser extent, of oxygen ($\delta^{18}$O), have been used in some studies to examine relationships between plants and their environment [20–22]. Although trees have been the primary focus of previous studies, grapevines were researched as well [23–25]. The carbon isotope ratio of the grape must is considered an integrated marker of the vine water status during berry growth, particularly during the period from veraison to harvest [24,26–28]. When vines experience water deficit, stomatal closure is enhanced, and, as a consequence, the relative proportion of $^{13}$C in the leaf increases because it is less reactive to the enzyme RuBisCO than $^{12}$C. Therefore, it is common to observe that plants growing under a water deficit exhibit greater $\delta^{13}$C than those that are not-water-stressed [29,30]. Regarding the oxygen isotopic ratio, it has recently been used as an indicator of transpiration in the days preceding harvest because of the high correlation found between the vapor pressure deficit (VPD) and $\delta^{18}$O under drought conditions [25]. In the present study, the carbon and oxygen isotope ratios were used to assess the vine water status and transpiration rate, respectively, during the berry-ripening period.

The aim of this work was to identify varieties responding well, in terms of yield and must quality, when subjected to severe water stress conditions for three consecutive years. For this purpose, five vines from 12 white grape varieties grown in the La Mancha wine region were monitored in a multivarietal vineyard. Agronomic indicators, such as phenology, vegetative development, yield components, and grape quality, were measured. The $\delta^{13}$C measurements provided a comprehensive estimate of the vines' water status and their efficiency in water use, helping to establish differences among varieties that would have been more difficult to perceive with point measurements. The use of varieties tolerant to drought may be an easy and effective way to ensure that viticulture remains sustainable in semi-arid climate areas in the near future.

## 2. Materials and Methods

### 2.1. Experimental Site and Plant Material

The study was conducted in an experimental vineyard located in the Regional Institute of Agri-Food and Forestry Research and Development of Castilla–La Mancha (IRIAF), in Tomelloso, Spain (latitude 39°10′34″ N, longitude 3°00′01″ W; altitude 660 m.a.s.l.). Six widely grown varieties and six minority varieties were selected. In this work, a minority variety is considered to be one whose growing area in the Castilla–La Mancha region is less than 1000 ha and whose growing area in the country as a whole does not exceed 3000 ha (Table 1).

In this work, which was conducted from 2018 to 2020, the varieties were arranged in parallel rows of 140 vines each. The vines were 15 years old and grafted onto 110-Richter rootstock. Additionally, they were trained on a bilateral Royat Cordon system and planted with a row spacing of 3 m and a vine spacing of 1.5 m (2222 vines ha$^{-1}$). The vines were pruned with six two-bud spurs per vine. Twenty-two contiguous vines per variety were monitored.

**Table 1.** List of 12 studied white grapevine varieties.

|  | Variety |
| --- | --- |
| Widely grown varieties | Airén |
|  | Jaén Blanco (*syn.* Pardina) |
|  | Macabeo (*syn.* Viura) |
|  | Moscatel de Grano Menudo |
|  | Pedro Ximénez |
|  | Verdejo |
| Minority varieties | Alarije (*syn.* Malvasía Riojana) |
|  | Albillo Real |
|  | Coloraillo |
|  | Malvar |
|  | Merseguera (*syn.* Exquitsagos) |
|  | Pardillo (*syn.* Marisancho) |

*2.2. Soil and Meteorological Data*

The soil of the vineyard was a Petric Calcisol (FAO soil classification) or Petrocalcic Calcixerept (USDA soil classification) with loam texture and active limestone and organic matter contents of 15% and 3%, respectively. The soil depth was 30 cm, below which there was a petrocalcic horizon, impenetrable by the vine roots. This type of soil, which is widely found throughout the La Mancha wine region, is traditionally associated with grapevine cultivation, and the pedoclimatic soil conditions are characterized by the typical xeric moisture regime of Mediterranean climates. The vineyard soil was managed mechanically by mowing the permanent natural plant cover throughout the year.

The climate is Mediterranean continental semi-arid, with hot and dry summers and cold and moderately rainy winters. Historical data from 2001 to 2020, recorded at the Argamasilla de Alba weather station, which is 12 km from the experimental site, were used to characterize the climate. The station belongs to the SIAR network of the Spanish Ministry for Agriculture, Fisheries, and Food. Throughout the study period, data on the mean temperatures and monthly rainfall were recorded (Table 2). The annual thermal amplitude is high, about 21.5 °C (Figure 1). The mean annual rainfall is approximately 380 mm, of which only about 40% falls during the grapevine growing season. Drought periods arelong (4.5 months). The Huglin index for the area is 2740, which classifies it as a warm climate ($HI_{+2}$) [31].

**Table 2.** Total rainfall and monthly mean temperature at the experimental site during the 2018, 2019, and 2020 agronomic years.

| Month | 2018 | | 2019 | | 2020 | |
| --- | --- | --- | --- | --- | --- | --- |
|  | Mean Temperature (°C) | Rainfall (mm) | Mean Temperature (°C) | Rainfall (mm) | Mean Temperature (°C) | Rainfall (mm) |
| Oct. | 17.2 | 20.0 | 14.6 | 39.3 | 16.5 | 18.0 |
| Nov. | 8.9 | 22.5 | 9.7 | 59.0 | 9.1 | 57.4 |
| Dec. | 4.6 | 33.5 | 6.1 | 11.8 | 7.6 | 36.0 |
| Jan. | 5.5 | 35.2 | 4.6 | 7.8 | 5.6 | 32.2 |
| Feb. | 4.4 | 74.3 | 7.9 | 10.6 | 9.9 | 4.4 |
| Mar. | 8.1 | 116.5 | 10.9 | 10.8 | 10.5 | 64.4 |
| Apr. | 12.3 | 37.6 | 11.5 | 124.0 | 13.1 | 52.6 |
| May | 15.9 | 39.8 | 18.0 | 19.2 | 19.1 | 37.6 |
| Jun. | 21.5 | 27.6 | 23.6 | 0.2 | 22.8 | 1.0 |
| Jul. | 25.8 | 0.0 | 27.5 | 7.6 | 28.0 | 13.3 |
| Aug. | 27.0 | 0.0 | 26.0 | 9.8 | 26.0 | 15.5 |
| Sep. | 22.6 | 0.0 | 21.0 | 33.2 | 20.7 | 6.0 |
| **Total** | **14.5** | **407.0** | **15.1** | **333.3** | **15.7** | **338.4** |

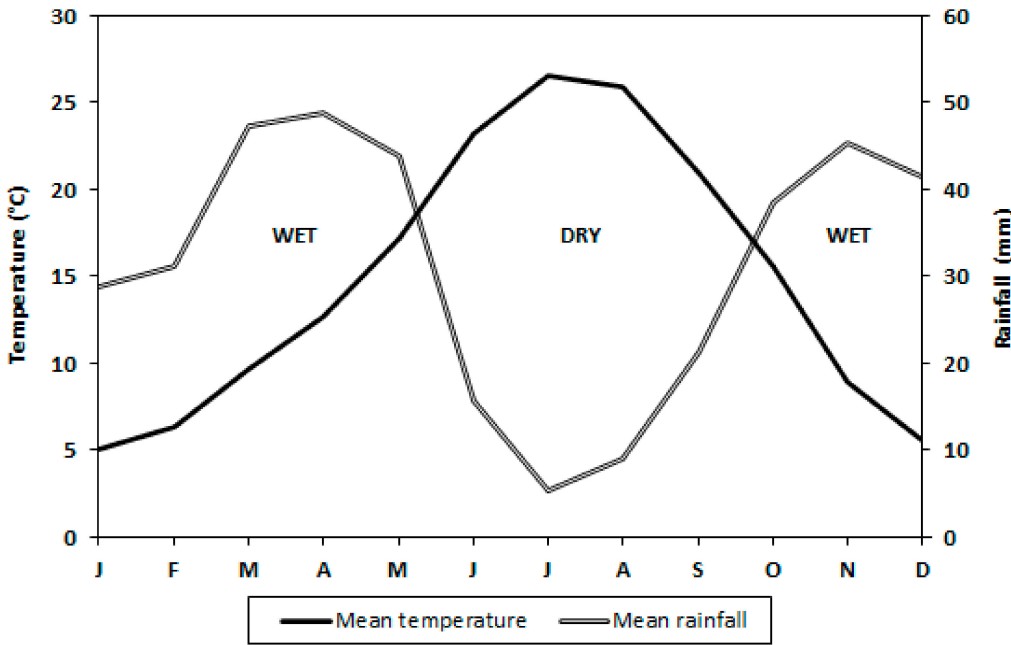

**Figure 1.** Climogram of the area based on climate historical data from 2001 to 2020.

### 2.3. Phenology

The phenology of the varieties was monitored, with the dates on which the main season phenological stages took place noted according to the BBCH scale [32]: budbreak (07), flowering (65), veraison (81), and maturity (89). The dates of the phenological stages were assigned when 50% of the buds or bunches of the monitored vines reached that stage, with the exception of maturity, on which the date was assigned when 100% of the grapes were mature. Figure 2 graphically represents the mean phenology stages of each variety.

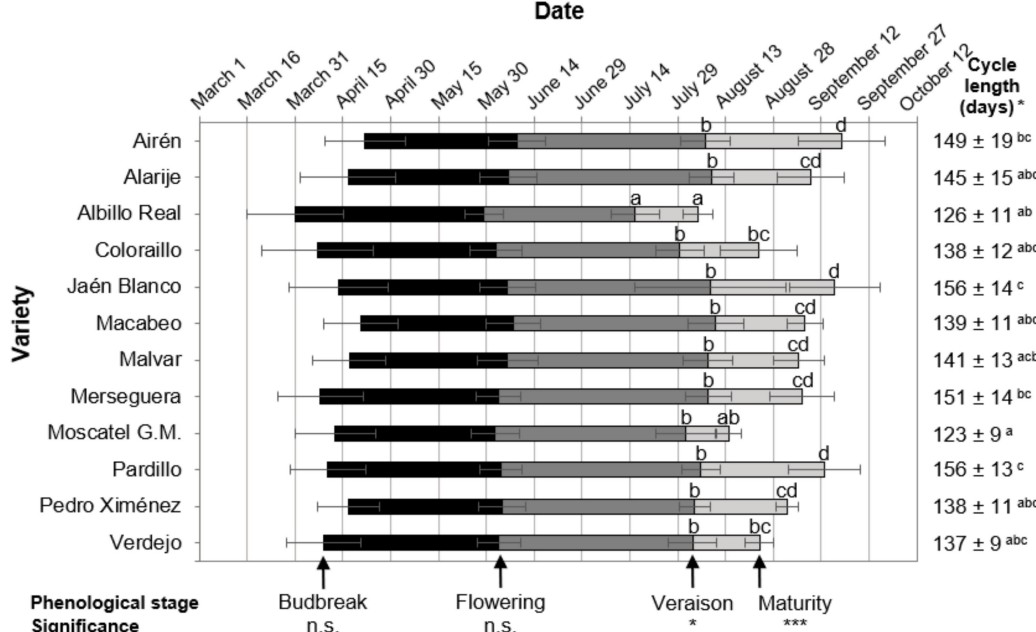

**Figure 2.** Length and date of each phenological stage in the different varieties. Bars are three-year means, and error bars are standard deviations. Different colored bars indicate the length of each period between phenological stages. Cycle length shows mean value ± standard deviation. Different letters indicate statistical differences among varieties by Duncan test (ANOVA; *, $p < 0.05$; ***, $p < 0.001$; n.s., not significant).

### 2.4. Water Regime

The vineyard was managed under rainfed conditions. However, to ensure grapevine survival, it was necessary to provide three irrigation doses of 10 mm each throughout the season each year (after fruit set, when the berries reached the size of a pea, and when the branches stopped growing). An on-surface drip irrigation system was used with drippers spaced at 0.75 m within the plant rows, with a nominal flow rate of 4 L h$^{-1}$.

### 2.5. Yield Components and Pruning Weight

To determine the harvest date, regular sampling was conducted until the grapes reached a concentration of total soluble solids between 20 and 22 °Brix. Five vines were harvested per variety. The following yield components were measured in each vine: yield (kg vine$^{-1}$), mean bunch weight (g), and mean berry weight (g). The mean bunch weight was obtained by dividing the yield by the number of bunches per vine. A sample of 100 randomly selected berries per vine was collected to measure the mean berry weight. The vines were pruned in winter, and the pruning weight (kg vine$^{-1}$) was assessed for each of the five vines monitored. To evaluate the balance between production and vigor, the Ravaz index was calculated for each vine.

### 2.6. Must Composition

The grapes of each harvested vine were individually pressed with a manual screw press to extract the must. Following the official methods of the International Organization of Vine and Wine [33], the total soluble solids of the must (°Brix) were measured by electronic refractometry (RX-5000$\alpha$-Bev, Atago, Tokyo, Japan), and the total acidity (g L$^{-1}$ tartaric acid) and pH were measured by potentiometry (HI 902, Hanna, Eibar, Spain). The must was sampled (12 mL per vine) and frozen in polycarbonate test tubes to allow for later isotopic composition analysis ($\delta^{13}$C and $\delta^{18}$O).

### 2.7. Carbon Isotopic Composition ($\delta^{13}$C)

The carbon isotope composition of the grape must was measured by on-line analysis using a ThermoQuest Flash 1112 Elemental Analyzer equipped with an autosampler and coupled to a Delta-Plus IRMS (ThermoQuest, Bremen, Germany) through a ConFlo III interface (ThermoQuest). One microliter of must was placed in a tin capsule and sealed. All of the carbon in the sample was oxidized to $CO_2$ by the reactors of the elemental analyzer. The analyzer passed the gas through a gas chromatography (GC) column to separate the $CO_2$ from the other gases and then brought the $CO_2$ into the mass spectrometer by a helium flow. The carbon isotope composition was expressed as:

$$\delta^{13}C_{sample} = [(R_s/R_{std}) - 1] \times 1000 \tag{1}$$

where $R_s$ is the $^{13}C/^{12}C$ ratio of the sample, and $R_{std}$ is the international reference standard Vienna Pee Dee Belemnite. Five vines of each variety were sampled, with two must samples per vine.

### 2.8. Oxygen Isotopic Composition ($\delta^{18}$O)

An on-line gas equilibration and headspace introduction system model, GasBench II (ThermoQuest, Bremen, Germany), was used to analyze the must oxygen isotopes. It was equipped with a GC column (PoraPlot Q, 25 m, 0.25 mm; Varian, Palo Alto, Santa Clara, CA, USA) operating at 70 °C and adapted to an autosampler CombiPAL (CTC-Analytics, Zwingen, Switzerland). For each must sample, 500 µL of must and a spatula tip of benzoic acid (to avoid possible fermentation) were transferred to a 10 mL vial with silicone septa. The vials were placed in the GasBench II, flushed with 0.3% $CO_2$ in He for 10 min, and left

for 48 h at 22 °C before analysis. During this equilibration time, an exchange reaction took place between the oxygen in the $CO_2$ and $H_2O$:

$$^{12}C^{16}O_2 + H_2{}^{18}O \leftrightarrow {}^{12}C^{16}O^{18}O + H_2{}^{16}O \tag{2}$$

The $CO_2$ was then isolated from the vial headspace and introduced in the IRMS system. The GasBench II was coupled to a Delta-Plus IRMS (ThermoQuest) equipped with three Faraday cup detectors that simultaneously and continuously monitored the $[CO_2]^+$ signals for the three major ions at $m/z$ 44 ($^{12}CO_2$), $m/z$ 45 ($^{13}CO_2$ and $^{12}C^{17}O^{16}O$), and $m/z$ 46 ($^{12}C^{18}O^{16}O$). The oxygen isotope composition was expressed as:

$$\delta^{18}O_{sample} = [(R_s/R_{std}) - 1] \times 1000 \tag{3}$$

where $R_s$ is the $^{18}O/^{16}O$ ratio of the sample, and $R_{std}$ is the international reference standard Vienna Standard Mean Ocean Water. Five vines of each variety were sampled, with two must samples per vine.

### 2.9. Statistical Analysis

First, the three-year data set ($n$ = 15) was analyzed for outliers (data exceeding three standard deviations). No outliers were found, meaning all data for each variable came from the same distribution. The data were analyzed using a one-way analysis of variance (ANOVA, $\alpha$ = 0.05), and the means were separated by Duncan's multiple range test using Statgraphics Centurion XVIII software (Statgraphics Technologies, The Plains, VA, USA). The effects of the year and variety, and the year–variety interaction were evaluated by two-way ANOVA. Principal component analysis (PCA) was performed to study differences and similarities among varieties in their agronomic behavior by Statgraphics software. The PCA groups were identified by cluster analysis. The bar graphs were created using SigmaPlot 14.0 software (Systat Software, San José, CA, USA).

## 3. Results

### 3.1. Phenology

The varieties did not differ significantly in their mean budbreak and flowering dates (Figure 2). However, the veraison, maturity, and cycle length demonstrated significant differences among varieties. Among all varieties, Albillo Real was the one with the earliest veraison and maturity dates, surpassing Moscatel de Grano Menudo. Both varieties exhibited a short cycle length of less than 130 days. Other early-maturing varieties were Coloraillo and Verdejo, which were harvested at the end of August. By contrast, Airén, Jaén Blanco, and Pardillo exhibited late ripening, with Jaén Blanco and Pardillo having the longest cycles of more than 155 days.

### 3.2. Yield Components and Pruning Weight

Figure 3 depicts the mean values of the yield components, pruning weight, and Ravaz index (yield to pruning weight ratio) obtained from the three years. All parameters indicate highly significant differences among the varieties ($p < 0.001$). Jaén Blanco and Merseguera were the varieties with the highest mean yields, 4.11 and 3.98 kg vine$^{-1}$, respectively, but for different reasons. In the case of Jaén Blanco, the high yield was determined by the berry and bunch weight, while in Merseguera, it was because of the high number of bunches. Conversely, Pedro Ximénez and Moscatel de Grano Menudo provided the lowest mean yields of 1.79 and 1.88 kg vine$^{-1}$, respectively. Coloraillo exhibited the highest pruning weight, with 0.71 kg vine$^{-1}$. By contrast, Pedro Ximénez had the lowest pruning weight, with 0.21 kg vine$^{-1}$. When the varieties were ranked by their berry weight, Airén and Jaén Blanco stood out above the other varieties, with weights greater than 2 g. Regarding the Ravaz index, measuring the yield and pruning weight, Merseguera stood out with the highest values (12.02), while Coloraillo exhibited the lowest (2.85).

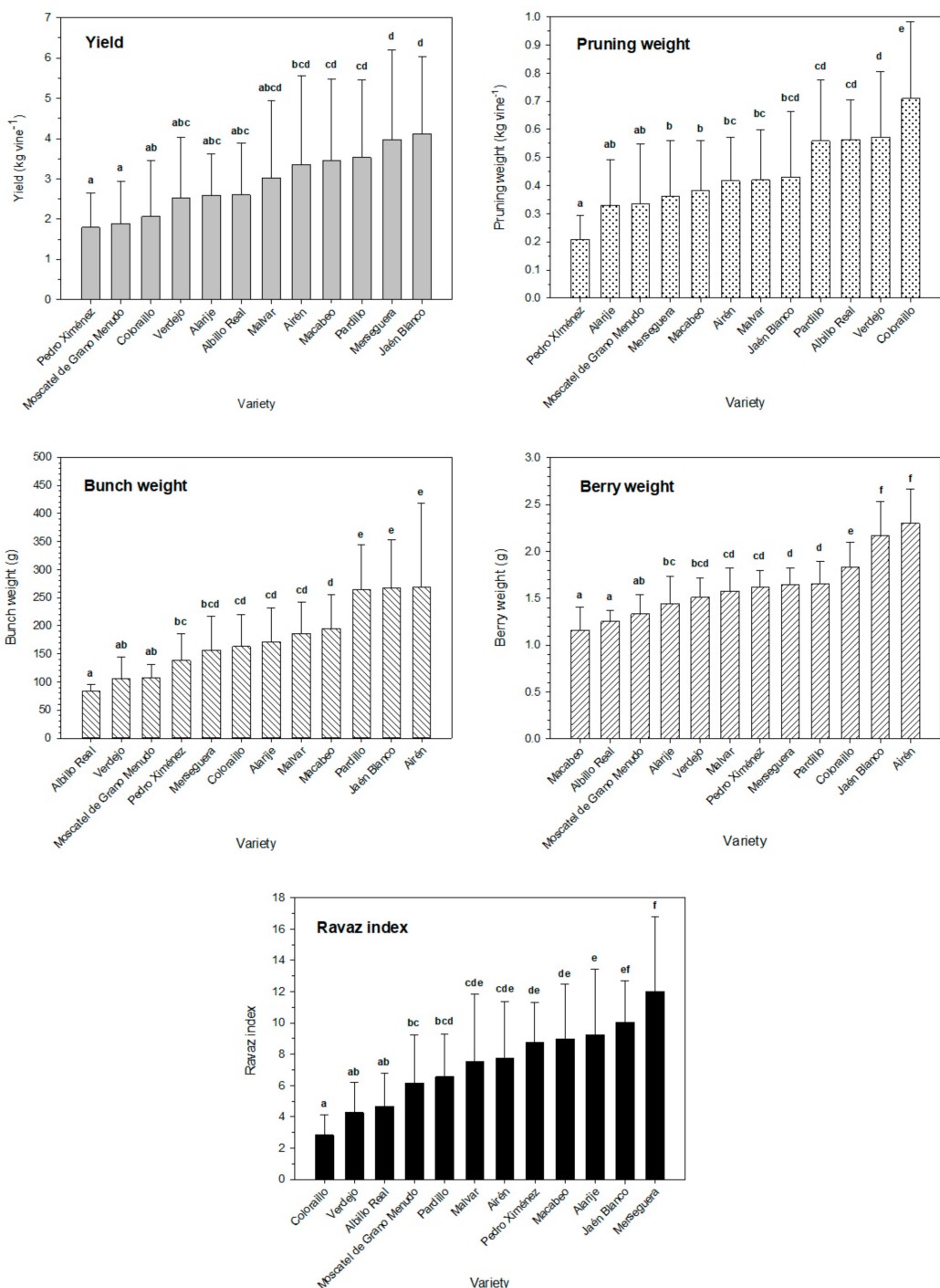

**Figure 3.** Yield components, pruning weight, and Ravaz index. Columns and error bars represent means and standard deviations, respectively, of 15 samples (five replicates each year). Different letters indicate statistical differences among varieties by the Duncan test (ANOVA, $p < 0.001$).

To determine the possible effects of the year, variety, and their interaction, a two-way analysis of variance was carried out with all parameters analyzed. Table 3 shows the results of the yield components, pruning weight, and Ravaz index. Both the year and the variety effect were highly significant ($p < 0.001$) for all parameters. The significance of the year–variety interaction regarding the yield and pruning weight was lower ($p < 0.05$) than in the case of the bunch and berry weight ($p < 0.001$), and there was no significance in the Ravaz index.

**Table 3.** Two-way ANOVA analysis of yield components, pruning weight, and Ravaz index.

| Two-Way ANOVA | Yield | Bunch Weight | Berry Weight | Pruning Weight | Ravaz Index |
|---|---|---|---|---|---|
| Year effect | $p < 0.001$ | $p < 0.001$ | $p < 0.001$ | $p < 0.001$ | $p < 0.001$ |
| Variety effect | $p < 0.001$ | $p < 0.001$ | $p < 0.001$ | $p < 0.001$ | $p < 0.001$ |
| Year–variety interaction | $p < 0.05$ | $p < 0.001$ | $p < 0.001$ | $p < 0.05$ | n.s. |

*3.3. Must Quality Parameters*

The must quality parameters displayed high variability among the varieties. Table 4 presents the must quality parameters' mean values for the different varieties, ranked in ascending order of total soluble solids. Pedro Ximénez exhibited the highest concentration of total soluble solids (22.15 °Brix), but it had a low total acidity concentration (3.57 g L$^{-1}$). Moscatel de Grano Menudo and Airén stood out for their high values of total acidity (7.55 g L$^{-1}$) and pH (3.54), respectively. The effects of year and variety were very significant on the total soluble solids, total acidity, and pH ($p < 0.001$). The effect was less significant ($p < 0.05$) on the must quality parameters when the year–variety interaction was considered.

**Table 4.** Must quality parameters (mean value and standard deviation, *n* = 15) of 12 white varieties.

| Variety | Total Soluble Solids ** (°Brix) | | Total Acidity *** (g L$^{-1}$) | | pH *** | |
|---|---|---|---|---|---|---|
| | Mean | SD | Mean | SD | Mean | SD |
| Merseguera | 19.68 [a] | 1.33 | 3.79 [ab] | 0.35 | 3.21 [a] | 0.18 |
| Alarije | 19.94 [ab] | 1.69 | 3.27 [a] | 0.53 | 3.43 [cde] | 0.18 |
| Airén | 20.00 [ab] | 1.58 | 3.42 [a] | 0.31 | 3.54 [e] | 0.21 |
| Pardillo | 20.17 [ab] | 1.15 | 3.38 [a] | 0.53 | 3.48 [de] | 0.15 |
| Malvar | 20.22 [ab] | 1.41 | 3.47 [a] | 0.27 | 3.38 [bcd] | 0.09 |
| Coloraillo | 20.28 [ab] | 2.09 | 5.80 [e] | 1.04 | 3.25 [ab] | 0.14 |
| Jaén Blanco | 20.56 [ab] | 1.87 | 4.13 [b] | 0.47 | 3.47 [de] | 0.25 |
| Macabeo | 20.77 [abc] | 2.31 | 4.61 [c] | 0.86 | 3.30 [ab] | 0.10 |
| Albillo Real | 20.99 [abc] | 1.85 | 5.12 [d] | 0.55 | 3.28 [ab] | 0.17 |
| Verdejo | 21.38 [bc] | 1.61 | 5.75 [e] | 0.60 | 3.33 [abc] | 0.16 |
| Moscatel de Grano Menudo | 21.48 [bc] | 1.61 | 7.55 [f] | 1.33 | 3.22 [a] | 0.21 |
| Pedro Ximénez | 22.15 [c] | 2.30 | 3.57 [a] | 0.59 | 3.45 [cde] | 0.15 |
| **Two-way ANOVA** | **Significance** | | | | | |
| Year effect | $p < 0.001$ | | $p < 0.001$ | | $p < 0.001$ | |
| Variety effect | $p < 0.001$ | | $p < 0.001$ | | $p < 0.001$ | |
| Year x variety interaction | $p < 0.05$ | | $p < 0.05$ | | $p < 0.05$ | |

Different letters in the same column denote statistically significant differences among varieties (ANOVA, Duncan test; **, $p < 0.01$; ***, $p < 0.001$).

*3.4. Must Carbon Isotope Ratio*

Table 5 presents the mean, minimum, and maximum values of the $\delta^{13}$C obtained in the must of the varieties from the 2018 to 2020 vintages. The $\delta^{13}$C values were highly similar, ranging from −23.445‰ to −22.729‰ (a 0.716‰ difference between the mean values). Even so, there were significant differences among the varieties that allowed for establishing different groups according to Duncan's test ($p < 0.01$). Jaén Blanco and Airén exhibited the highest mean $\delta^{13}$C values, indicating that they were the most efficient varieties in water use, whereas Albillo Real and Pardillo displayed low $\delta^{13}$C values and, thus, were less efficient. Other varieties, such as Merseguera, Verdejo, and Pedro Ximénez, were classified as moderately efficient. Albillo Real (3.960‰) and Verdejo (1.382‰) were the varieties with the highest and lowest intra-variety variabilities in $\delta^{13}$C, respectively, in the three years of

study. The year and variety effects were more significant on the $\delta^{13}$C ($p < 0.001$), than the year–variety interaction ($p < 0.01$).

**Table 5.** Carbon isotope ratio measured at harvest date in the must sugar of 12 different white varieties. Values are means, standard deviation, minimum, and maximum values of 15 must samples (five replicates each year).

| Variety | $\delta^{13}$C (‰) | | | |
|---|---|---|---|---|
| | **Mean** | **SD** | **Min** | **Max** |
| Albillo Real | −23.445 [a] | 0.882 | −25.519 | −21.559 |
| Pardillo | −23.424 [a] | 0.835 | −24.527 | −22.200 |
| Macabeo | −23.351 [ab] | 0.511 | −24.558 | −22.420 |
| Coloraillo | −23.315 [abc] | 0.552 | −24.347 | −22.581 |
| Merseguera | −23.202 [abcd] | 0.594 | −24.100 | −22.149 |
| Verdejo | −23.140 [abcd] | 0.364 | −23.969 | −22.587 |
| Pedro Ximénez | −23.123 [abcd] | 0.454 | −23.899 | −22.168 |
| Moscatel de Grano Menudo | −22.930 [bcd] | 0.609 | −23.834 | −21.793 |
| Alarije | −22.883 [bcd] | 0.428 | −23.658 | −22.150 |
| Malvar | −22.847 [cd] | 0.390 | −23.662 | −22.204 |
| Airén | −22.734 [d] | 0.674 | −24.123 | −21.720 |
| Jaén Blanco | −22.729 [d] | 0.724 | −24.040 | −21.793 |
| **Two-way ANOVA** | **Significance** | | | |
| Year effect | $p < 0.001$ | | | |
| Variety effect | $p < 0.001$ | | | |
| Year–variety interaction | $p < 0.01$ | | | |

Different letters in the same column denote statistically significant differences among varieties (ANOVA, Duncan test, $p < 0.01$).

### 3.5. Must Oxygen Isotope Ratio

The oxygen isotope ratio of the grape must water was assessed to differentiate the varieties according to the water losses caused by transpiration during the seven days prior to the harvest date (Table 6). The $\delta^{18}$O ranged widely among the varieties (3.072‰), and the differences were highly significant ($p < 0.001$). Merseguera and Alarije exhibited the lowest mean $\delta^{18}$O values of 9.026‰ and 9.266‰, respectively, indicating that these varieties had the lowest transpiration rates during the last days of ripening, while Albillo Real exhibited the highest mean (12.098‰). The varieties with the highest and lowest intra-variety variabilities during the three years were Merseguera (5.788‰) and Albillo Real (1.515‰), respectively. Similar to $\delta^{13}$C, the year and variety effects were more significant on $\delta^{13}$O ($p < 0.001$), than the year–variety interaction ($p < 0.01$).

### 3.6. Analysis of the Agronomic Behavior of the Varieties Using PCA

Taking into account the main variables measured—yield, berry weight, pruning weight, total soluble solids, total acidity, pH, and $\delta^{13}$C—a multivariate data analysis using PCA was performed to determine how these traits changed each year among different varieties when they were grown under drought conditions (Figure 4). Figure 4 shows the distribution of the varieties considering each year individually and jointly regarding these variables in the space defined by the PCA. As expected, both the effect of the environmental conditions during the vintage and the variety significantly influenced the results. However, the most relevant results are due to the way in which the different varieties were grouped and their distribution in the PCA analysis regarding the different parameters analyzed, which means that they responded in a similar way to water stress. During the three years of the study, the distribution of the varieties was analogous. Albillo Real, Coloraillo, Macabeo, and Verdejo were placed close to the axes corresponding to the total acidity and pruning weight. By contrast, varieties such as Airén, Alarije, Jaén Blanco, Malvar, Pardillo, and Merseguera approached the yield, berry weight, pH, and $\delta^{13}$C axes. These findings support

the argument of using the mean values of the parameters analyzed considering the three years in the PCA analysis to evaluate the response of varieties to drought. According to the PCA performed with the mean values, the first two principal components explained 65.43% of all variance for the 12 varieties (42.34 and 23.09% for F1 and F2, respectively) and the cluster analysis enabled the varieties to be separated into five groups.

**Table 6.** Oxygen isotope ratio measured at harvest date in the must water of 12 different white varieties. Values are means, standard deviation, minimum, and maximum values of 15 must samples (five replicates each year).

| Variety | $\delta^{18}O$ (‰) | | | |
|---|---|---|---|---|
| | Mean | SD | Min | Max |
| Merseguera | 9.026 [a] | 1.833 | 6.324 | 12.112 |
| Alarije | 9.266 [a] | 1.563 | 6.821 | 11.881 |
| Malvar | 9.354 [ab] | 1.927 | 6.677 | 12.384 |
| Macabeo | 9.386 [ab] | 1.634 | 6.813 | 11.751 |
| Pardillo | 9.551 [ab] | 0.846 | 8.144 | 10.964 |
| Jaén Blanco | 9.709 [abc] | 1.162 | 8.336 | 11.955 |
| Verdejo | 10.356 [bc] | 1.398 | 8.328 | 12.639 |
| Airén | 10.629 [cd] | 0.647 | 9.501 | 12.040 |
| Pedro Ximénez | 10.642 [cd] | 1.689 | 8.328 | 13.624 |
| Moscatel de Grano Menudo | 10.694 [cd] | 1.243 | 8.372 | 12.299 |
| Coloraillo | 11.623 [de] | 0.995 | 10.240 | 13.163 |
| Albillo Real | 12.098 [e] | 0.413 | 11.150 | 12.665 |

| Two-way ANOVA | Significance |
|---|---|
| Year effect | $p < 0.001$ |
| Variety effect | $p < 0.001$ |
| Year–variety interaction | $p < 0.01$ |

Different letters in the same column denote statistically significant differences among varieties (ANOVA, Duncan test, $p < 0.001$).

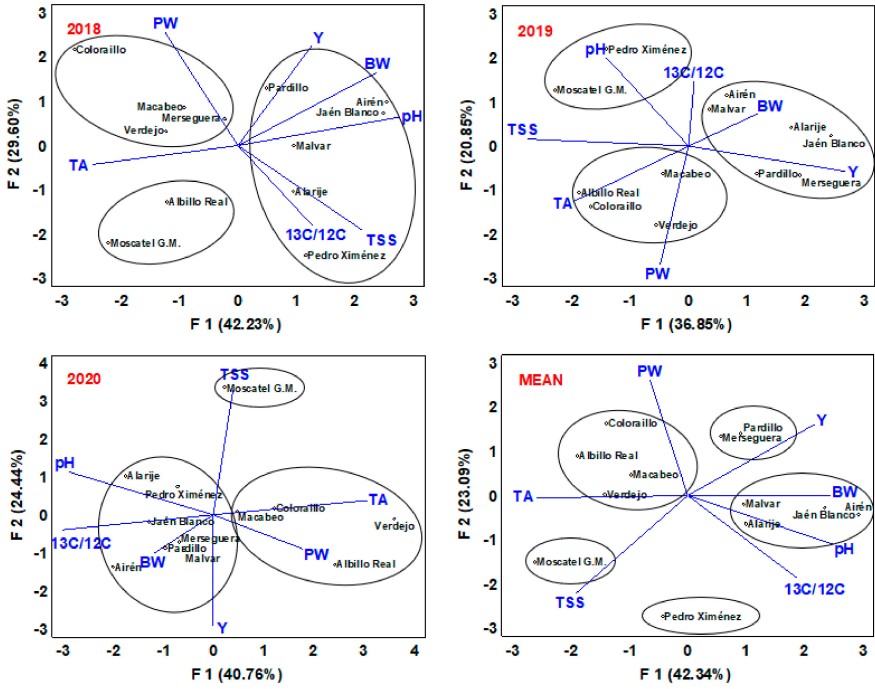

**Figure 4.** Principal components analysis (PCA) considering the data of each year separately and jointly for the triennium 2018–2020. Y, yield; BW, berry weight; PW, pruning weight; TSS, total soluble solids; TA, total acidity; pH; 13C/12C, carbon isotope ratio.

## 4. Discussion

The response to severe water stress ranged widely among the varieties in terms of yield, must quality, and water-use efficiency. The effects of interannual climatic variability and variety were highly significant in all parameters. By contrast, the year–variety interaction was generally less significant. Even so, during the three years of the study, the varieties maintained their trends in terms of their positioning regarding the axes of the parameters analyzed. In general, vigorous and moderately productive varieties revealed the highest must quality. By contrast, these varieties exhibited low levels of water-use efficiency (WUE).

### 4.1. Productive Response

Previous studies have demonstrated that a yield-to-pruning weight ratio of 5/10 is an indicator of balanced vines capable of producing high-quality fruit [34–36]. According to this relationship, the vines of the Merseguera and Jaén Blanco varieties, with a Ravaz index greater than 10, were unbalanced due to excess yield compared to the pruning weight. Conversely, Albillo Real, Coloraillo, and Verdejo exhibited Ravaz indices below 5, meaning they were equally unbalanced, with vines of high vigor and low yields. These findings for the Verdejo variety partially agree with those obtained in another previous study for the same variety grown under rainfed conditions [37]. In our study, Verdejo exhibited similar yield and pruning weight values but a significantly lower bunch weight. These differences may have been due to soil or crop management. The effects of the interaction between the year and variety were not significant in the Ravax index, which means that regardless of the year, the yield-to-pruning weight ratio was maintained in the varieties.

The berry size is also an essential value to take into account in grapes, since the higher the skin-to-berry ratio, the greater the potential quality of the compounds synthesized in the skin. Particularly interesting in white grapes are the volatile compounds [38] that can be released into the wine. Therefore, varieties with large grapes, such as Airén and Jaén Blanco (above 2 g) are considered to theoretically have low potential quality. By contrast, Macabeo and Albillo Real, with moderate yields and low berry weight values, exhibited must quality parameters categorized from medium to high. In Figure 3, some high standard deviations—corresponding to a coefficient of variation of up to 60%—among the three years were observed for several varieties in terms of yield, bunch weight (Airén), and pruning weight (Jaén Blanco, Merseguera, and Moscatel de Grano Menudo). These findings are consistent with previously reported values for different varieties grown under water-deficit conditions, whose coefficients of variation regarding the yield reached 42% in Chile [39] and 50% in Italy [40].

### 4.2. Must Quality Response

In all varieties, the grapes ripened properly, but those of Pedro Ximénez stood out for their higher concentration of total soluble solids (above 22 °Brix). Musts from varieties such as Airén, Alarije, Malvar, Merseguera, Pardillo, and Pedro Ximénez exhibited low mean total acidity values (below 4 g $L^{-1}$), a detrimental feature to quality, particularly in white varieties. Conversely, the Moscatel de Grano Menudo must exhibited the highest mean total acidity value (above 7.5 g $L^{-1}$). The mean total acidity values in the musts of Verdejo (5.75 g $L^{-1}$) observed in this study are comparable to those obtained for this variety in another geographic area under similar experimental conditions [37].

### 4.3. Must Isotope Ratios

The values of $\delta^{13}C$ from the musts suggest severe drought stress in all varieties, with mean values above −24‰ [24]. These findings are consistent with previously reported values for the same location under a non-irrigated regime [23,25]. Significant variation among the varieties in the $\delta^{13}C$ was found. According to the results, late-ripening varieties traditionally grown in the area, such as Jaén Blanco and Airén, exhibited high water-use efficiency, whereas Albillo Real, Pardillo, and Macabeo behaved as low-efficiency varieties. Our findings are consistent with those previously reported in the Balearic Islands by [41],

who also categorized Macabeo as a low-efficiency variety (low foliar $\delta^{13}$C). By contrast, the findings of this study partially disagree with obtained those by [23] in the La Mancha region, who found small differences regarding the $\delta^{13}$C values for Macabeo and Airén, therefore indicating that both varieties would reveal a similar water-use efficiency. These differences could be mainly due to different environmental conditions during the growing season in which the assays were performed, the characteristics of the rootstock traits, and the vine organ sampled.

Previous research has suggested that $\delta^{18}$O can complement $\delta^{13}$C in estimating the accumulated water deficit in vines under drought conditions [24]. Conversely, in assays performed on the wood of forest trees, the correlation found between the two isotope ratios has generally been poor [42]. In grapevines, it has been reported that the higher the canopy–air vapor pressure gradient, the larger the heavy oxygen isotope ($^{18}$O) content of the must water, making it a significant factor in determining the degree of must water enrichment under water deficit conditions [43], particularly during the seven days prior to harvest [25].

In our study, the highest $\delta^{18}$O value in must water was obtained in Albillo Real, a very early harvest variety, whose grapes ripened under atmospheric conditions with high daily mean VPD values. By contrast, in moderately late to late-harvest varieties—such as Merseguera, Alarije, Malvar, Macabeo, Pardillo, and Jaén Blanco—the grapes ripened on days when VPD values were low and, therefore, these varieties exhibited also lower $\delta^{18}$O values. Macabeo was harvested before Airén, but its $\delta^{18}$O values were significantly lower. These findings are partially consistent with the results obtained by other authors for these varieties, who, in the first year, obtained slightly lower $\delta^{18}$O values for Macabeo than for Airén. However, these values were significantly higher in the second year [23]. These variances may be due to differences in the harvest date and/or environmental conditions during the ripening stage.

*4.4. Synthesis of Results Classifying Varieties under Severe Water Stress Conditions Based on Their Agronomic Behavior*

In addition to the PCA analysis (see preliminary remarks in Section 3.6) and to synthesize the results obtained, a table was prepared, classifying the varieties into three arbitrary categories—good, medium, and poor—depending on the variety's response to severe water stress and using the mean values of the same variables chosen for the PCA (Table 7). As expected, there were no varieties with all traits classified in Category A. The first group—formed by Albillo Real, Coloraillo, Macabeo, and Verdejo—was characterized because, except for $\delta^{13}$C, all the other variables classified them in A or B categories. These varieties exhibited low to moderate efficiency in water use. By contrast, they performed well in terms of yield and quality grapes and, therefore, they can be considered suitable varieties to be grown under water stress conditions. In addition, these same varieties were grouped together in the 2019 and 2020 PCAs. The second group consisted of Pardillo and Merseguera, which exhibited a good response in terms of yield and pruning weight. However, both varieties indicated low concentrations of total acidity, which negatively affected their must quality. The last group was composed of Airén, Alarije, Jaén Blanco, and Malvar. These varieties were gathered in the same group in the 2018, 2019, and 2020 PCAs, and all four varieties were classified in the high or moderate categories for water-use efficiency and yield. Conversely, they exhibited a poor response in terms of must quality parameters and, therefore, they can be considered unsuitable to be grown under drought conditions. Finally, Moscatel de Grano Menudo and Pedro Ximénez were separately isolated at the bottom of the graph. Both varieties revealed similarities, such as a high concentration of total soluble solids and low yield and pruning weight—a trend that continued for all three years in the PCAs. By contrast, the quality of the Moscatel de Grano Menudo musts was higher than that of the Pedro Ximénez must because they had greater acidity.

**Table 7.** Classification of varieties into three categories based on their response to drought according to the seven parameters considered in the PCA: yield, berry weight, pruning weight, total soluble solids, total acidity, pH, and $\delta^{13}$C. Category cut-offs were arbitrarily chosen (adapted from [41]).

| Trait | Category A (Good) | Category B (Medium) | Category C (Poor) |
|---|---|---|---|
| Yield | (>3.5 kg vine$^{-1}$) Jaén Blanco, Merseguera, Pardillo | (2.0–3.5 kg vine$^{-1}$) Airén, Alarije, Albillo Real, Coloraillo, Macabeo, Malvar, Verdejo | (<2.0 kg vine$^{-1}$) Moscatel de Grano Menudo, Pedro Ximénez |
| Berry weight | (<1.5 g) Alarije, Albillo Real, Macabeo, Moscatel de Grano Menudo | (1.5–2 g) Coloraillo, Malvar, Merseguera, Pardillo, Pedro Ximénez, Verdejo | (>2 g) Airén, Jaén Blanco |
| Pruning weight | (>0.50 kg vine$^{-1}$) Albillo Real, Coloraillo, Pardillo, Verdejo | (0.35–0.50 kg vine$^{-1}$) Airén, Jaén Blanco, Macabeo, Malvar, Merseguera | (<0.35 kg vine$^{-1}$) Alarije, Moscatel de Grano Menudo, Pedro Ximénez |
| Total soluble solids | (>21.0 °Brix) Moscatel de Grano Menudo, Pedro Ximénez, Verdejo | (20.0–21.0 °Brix) Airén, Albillo Real, Coloraillo, Jaén Blanco, Macabeo, Malvar, Pardillo | (<20.0 °Brix) Alarije, Merseguera |
| Total acidity | (>5.5 g L$^{-1}$) Coloraillo, Moscatel de Grano Menudo, Verdejo | (4.0–5.5 g L$^{-1}$) Albillo Real, Jaén Blanco, Macabeo | (<4.0 g L$^{-1}$) Airén, Alarije, Malvar, Merseguera, Pardillo, Pedro Ximénez |
| pH | (<3.3) Albillo Real, Coloraillo, Macabeo, Merseguera, Moscatel de Grano Menudo | (3.3–3.5) Alarije, Jaén Blanco, Malvar, Pardillo, Pedro Ximénez, Verdejo | (>3.5) Airén |
| $\delta^{13}$C | (>−22.8‰) Airén, Jaén Blanco | (−22.8 to −23.3‰) Alarije, Malvar, Merseguera, Moscatel de Grano Menudo, Pedro Ximénez, Verdejo | (<−23.3‰) Albillo Real, Coloraillo, Macabeo, Pardillo |

## 5. Conclusions

This study revealed high variability in the behavior of grapevine varieties when the water supply is dramatically reduced. Agronomic factors such as the yield, pruning weight, must quality parameters, and water-use efficiency were significantly influenced by variety. The results confirm a group of four varieties as candidates to be grown under severe water stress conditions: Albillo Real, Coloraillo, Macabeo, and Verdejo. These varieties behaved inefficiently in their water use, but they were able to maintain a balance between the yield and must quality. This behavior suggests that they may be suitable for cultivation in semi-arid climatic regions with limited water resources. Conversely, the efficiency in the water use of Airén, the most commonly cultivated variety in the area, did not translate into improved must quality.

**Author Contributions:** Conceptualization and data collection, J.M.-G. and A.S.S.; methodology, J.M.-G., A.S.S. and J.L.C.-V.; software, A.S.S.; validation, J.M.-G. and J.L.C.-V.; formal analysis, A.S.S., M.D.C., A.M.M. and C.C.-T.; investigation, A.S.S., J.M.-G. and J.L.C.-V.; resources, G.L.A.; data curation, A.S.S.; writing—original draft preparation, A.S.S. and J.L.C.-V.; writing—review and editing, A.S.S., G.L.A., J.M.-G. and J.L.C.-V.; visualization, A.S.S., J.M.-G. and J.L.C.-V.; supervision, J.M.-G., A.S.S. and J.L.C.-V.; project administration, G.L.A.; funding acquisition, G.L.A. All authors have read and agreed to the published version of the manuscript.

**Funding:** This research received no external funding.

**Data Availability Statement:** Not applicable.

**Acknowledgments:** A. Sergio Serrano is grateful for her predoctoral contract granted by the University of Castilla-La Mancha (UCLM) and co-financed by the European Social Fund under the Operational Programme 2014–2020 of Castilla–La Mancha.

**Conflicts of Interest:** The authors declare no conflict of interest.

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
