# Peer review of "Variability in the Agronomic Behavior of 12 White Grapevine Varieties Grown under Severe Water Stress Conditions in the La Mancha Wine Region"

_horticulturae, doi:10.3390/horticulturae9020243_

Round 1

Reviewer 1 Report

The manuscript entitled “Variability in the agronomic behavior of 12 white grapevine varieties grown under severe water stress conditions in the La Mancha wine region” addresses the identification of grapevine varieties that respond well, in terms of yield and must quality, when subjected to severe water stress conditions for three consecutive years. The manuscript seems to be well-designed and written and could be very suitable to increase the knowledge about the best strategy for wine-growing regions with increasingly severe arid conditions. This topic is part of one of the most important long-term adaptation measures to climate change that the scientific community should develop more intensively, in order to provide very robust guidelines to stakeholders. I support and recommend the publication of this study. However, there are some methodological inaccuracies that must be reviewed. I suggest that they be considered in order to improve the quality of the article. More accurately, I have the following specific comments:

1)     Statistical analysis: The study was done over 3 years. Very well. However, given that the 3 years were meteorologically distinct, this effect must also be considered in the statistical analysis. Please, the data must be analysed using a two-way analysis of variance (year effect, cultivar effect and respective interaction). As the data are treated, the analysis is too simplistic and hides the interannual variability, which may lead to inappropriate conclusions.

2)     The authors refer that this study was carried out in severe water stress conditions. However, no plant water indicator is presented to support this statement.

3)     Water regime (lines 131-135): “However, to ensure grapevine survival, it was necessary to provide three irrigation doses of 10 mm each throughout the season each year” – given the rainfall that occurred in the summer of 2019 and 2020, was it necessary to irrigate the same amount as in 2018? Is this water management correct?

4)     Legend of table 2: “Total rainfall and …” – Only monthly precipitation values are shown. Total rainfall from October to September must also be shown.

5)     All cultivars were pruned with six two-bud spurs per vine, leading to large differences in yield, vigour and quality attributes. Is this strategy correct? In the discussion section, this practice should be discussed.

6)     Line 289-290: “In Figure 3, high coefficients of variation (CV) between the three years …” –Please, rewrite this sentence or introduce this statistical element in Figure 3.

7)     Title: I think the word "behavior" should be written according to British English rules...

Author Response

Dear reviewer,

The authors thank you very much for your valuable comments. These were considered in the preparation of the new manuscript and the text was modified. Regarding your comments, the following were corrected as you recommended.

1)         Statistical analysis: The study was done over 3 years. Very well. However, given that the 3 years were meteorologically distinct, this effect must also be considered in the statistical analysis. Please, the data must be analysed using a two-way analysis of variance (year effect, cultivar effect and respective interaction). As the data are treated, the analysis is too simplistic and hides the interannual variability, which may lead to inappropriate conclusions.

Thank you very much for your suggestion. The data has been analysed using a two-way analysis of variance and considering year effect, cultivar effect, and respective interaction as you recommended. The results are displayed at the bottom of each table in Results section. In addition, some comments have been added to manuscript and Figure 4 has been modified supporting the argument for using the mean values of the parameters in the PCA analysis.

2)     The authors refer that this study was carried out in severe water stress conditions. However, no plant water indicator is presented to support this statement.

The fact that it was necessary to provide three small irrigation doses throughout the cycle to ensure grapevine survival gives an idea of the severe water stress conditions in which the vines were grown. These severe conditions were subsequently verified with the measurements of the δ13C in grape musts, whose mean values were higher than -24‰ in all varieties.

3)     Water regime (lines 131-135): “However, to ensure grapevine survival, it was necessary to provide three irrigation doses of 10 mm each throughout the season each year” – given the rainfall that occurred in the summer of 2019 and 2020, was it necessary to irrigate the same amount as in 2018? Is this water management correct?

There are two reasons why it was decided to provide three doses of irrigation all years. The first was so that the amount of water supplied annually would be identical and in the same periods. The second was due to the fact that the rain that occurred in the summer of 2019 and 2020 took place separately on several and very hot days and as a consequence, the rainfall was little or nothing effective.

4)     Legend of table 2: “Total rainfall and …” – Only monthly precipitation values are shown. Total rainfall from October to September must also be shown.

Thank you for your observation. The total rainfall values are now shown in table 2.

5)     All cultivars were pruned with six two-bud spurs per vine, leading to large differences in yield, vigour and quality attributes. Is this strategy correct? In the discussion section, this practice should be discussed.

This type of pruning is usual in the area where the study was carried out. With this sentence we want to emphasize that all varieties were pruned in the same way. The aim was that the starting point was identical in all the varieties and, as a consequence, the results obtained in terms of yield, vigour and quality parameters could be comparable between them.

6)     Line 289-290: “In Figure 3, high coefficients of variation (CV) between the three years …” –Please, rewrite this sentence or introduce this statistical element in Figure 3.

Thank you for your observation. The sentence has been rewritten as you recommended.

7)     Title: I think the word "behavior" should be written according to British English rules...

With all due respect, given that the entire text was written according to the rules of American English, the same was true for the word 'behavior'. In addition, we consider that this denomination is more generalized in the subject we are working on.

Reviewer 2 Report

The manuscript is very interesting and easy to read. The findings are well presented and make a solid ground for the conclusions.

The only major remark is the section: 4.4. Classifying varieties based on their agronomic behavior under water stress conditions by 340 PCA analysis which have to be moved to the Results, not to be inserted in Discussion section.

Author Response

Dear reviewer,

The authors thank you very much for your valuable comment. It was considered in the preparation of the new manuscript. The text was modified as you recommended.

The only major remark is the section: 4.4. Classifying varieties based on their agronomic behavior under water stress conditions by 340 PCA analysis which have to be moved to the Results, not to be inserted in Discussion section.

Thank you for your observation. A part of the heading 4.4. Classifying varieties based on their agronomic behavior under water stress conditions by PCA analysis has been moved to the Results section and the title has been changed to 3.6. Analysis of the agronomic behavior of the varieties using PCA. The other part has remained in the Discussion section with another title: 4.4. Synthesis of results classifying varieties under severe water stress conditions based on their agronomic behavior. We think that Table 7 should remain at the end of the discussion section because it is the one that summarizes the results obtained in this paper.

Round 2

Reviewer 1 Report

The authors addressed all the issues raised and improved the clarity and quality of the manuscript. I agree absolutely with the small changes that were introduced in the manuscript. Therefore, it can be published in Horticulture in its current form.